# Retrospective Analysis of Six Years of Acute Flaccid Paralysis Surveillance and Polio Vaccine Coverage Reported by Italy, Serbia, Bosnia and Herzegovina, Montenegro, Bulgaria, Kosovo, Albania, North Macedonia, Malta, and Greece

**DOI:** 10.3390/vaccines10010044

**Published:** 2021-12-30

**Authors:** Stefano Fontana, Gabriele Buttinelli, Stefano Fiore, Concetta Amato, Marco Pataracchia, Majlinda Kota, Jela Aćimović, Mia Blažević, Mirsada Mulaomerović, Lubomira Nikolaeva-Glomb, Andreas Mentis, Androniki Voulgari-Kokota, Luljeta Gashi, Pranvera Kaçaniku-Gunga, Christopher Barbara, Jackie Melillo, Jelena Protic, Svetlana Filipović-Vignjevic, Patrick M. O’Connor, Alessandra D’Alberto, Riccardo Orioli, Andrea Siddu, Eugene Saxentoff, Paola Stefanelli

**Affiliations:** 1Department of Infectious Disease, Istituto Superiore di Sanità, 00161 Rome, Italy; stefano.fontana@iss.it (S.F.); gabriele.buttinelli@iss.it (G.B.); stefano.fiore@iss.it (S.F.); concetta.amato@iss.it (C.A.); marco.pataracchia@iss.it (M.P.); 2Laboratory of Virology, Department of Control of Infectious Diseases, Institute of Public Health, 1001 Tirana, Albania; mdhimolea@live.com; 3Department of Epidemiology, Public Health Institute of the Republic of Srpska, 78000 Banja Luka, Bosnia and Herzegovina; jela.acimovic@phi.rs.ba; 4Institute for Public Health of Federation Bosnia and Herzegovina, 71000 Sarajevo, Bosnia and Herzegovina; m.blazevic@zzjzfbih.ba (M.B.); mirsada.mulaomerovic@gmail.com (M.M.); 5Department of Virology, National Centre of Infectious and Parasitic Diseases, 1504 Sofia, Bulgaria; lubomira@gmail.com; 6National Poliovirus/Enterovirus Reference Laboratory, Diagnostic Department, Hellenic Pasteur Institute, 11521 Athens, Greece; mentis@pasteur.gr (A.M.); avoulgari@pasteur.gr (A.V.-K.); 7Department of Epidemiology, National Institute of Public Health, 10000 Pristina, Kosovo; luljetagashi02@yahoo.com (L.G.); pranveragunga@hotmail.com (P.K.-G.); 8Department of Pathology, Mater Dei Hospital, MSD2090 Msida, Malta; christopher.barbara@gov.mt; 9Department for Health Regulation, Health Promotion and Disease Prevention, MSD2090 Msida, Malta; jackie.m.melillo@gov.mt; 10National Reference Laboratory for ARBO Viruses and Hemorrhagic Fever, Institute of Virology, Vaccines and Sera “Torlak”, 11152 Belgrade, Serbia; jprotic@torlak.rs; 11Diagnostics and Research and Development, Institute of Virology, Vaccines and Sera “Torlak”, 11152 Belgrade, Serbia; sfilipovic@torlak.rs; 12Global Immunization Division US Centers for Disease Control and Prevention, Atlanta, GA 30333, USA; GYP8@cdc.gov; 13Prevention of Communicable Diseases and International Prophylaxis, Directorate General of Health Prevention, Ministry of Health, 00144 Rome, Italy; a.dalberto@sanita.it (A.D.); r.orioli@sanita.it (R.O.); a.siddu@sanita.it (A.S.); 14Division of Health Emergencies and Communicable Diseases (DEC), Regional Office for Europe World Health Organization, DK-2100 Copenhagen, Denmark; saxentoffe@who.int

**Keywords:** enterovirus, poliovirus, non-polio enteroviruses, acute flaccid paralysis, poliovirus surveillance, polio vaccine coverage

## Abstract

Here we analyzed six years of acute flaccid paralysis (AFP) surveillance, from 2015 to 2020, of 10 countries linked to the WHO Regional Reference Laboratory, at the Istituto Superiore di Sanità, Italy. The analysis also comprises the polio vaccine coverage available (2015–2019) and enterovirus (EV) identification and typing data. Centralized Information System for Infectious Diseases and Laboratory Data Management System databases were used to obtain data on AFP indicators and laboratory performance and countries’ vaccine coverage from 2015 to 2019. EV isolation, identification, and typing were performed by each country according to WHO protocols. Overall, a general AFP underreporting was observed. Non-Polio Enterovirus (NPEV) typing showed a high heterogeneity: over the years, several genotypes of coxsackievirus and echovirus have been identified. The polio vaccine coverage, for the data available, differs among countries. This evaluation allows for the collection, for the first time, of data from the countries of the Balkan area regarding AFP surveillance and polio vaccine coverage. The need, for some countries, to enhance the surveillance systems and to promote the polio vaccine uptake, in order to maintain the polio-free status, is evident.

## 1. Introduction

Human Enteroviruses (EVs), including polioviruses (PVs), are non-enveloped, positive-sense single-stranded RNA viruses, transmitted via the fecal-oral route and respiratory tract. They are classified into four species (A, B, C, and D), with an increasing number of genotypes [1]. Although most EVs infections are asymptomatic, mild or severe diseases are reported in infants and in young children, such as self-limiting febrile illnesses, encephalitis, meningitis, acute flaccid paralysis (AFP), pancreatitis, myocarditis, encephalitis, and foot and mouth disease [1]. Among non-polio enteroviruses (NPEVs), EV-D68, EV-A71, and Echovirus genotypes can occur as an outbreak of infections with a relevant impact on public health [2,3,4]. EV-D68 in children can require intensive care support due to respiratory paralysis and a recent study reported epidemiological and biological evidence linking enterovirus D68 and acute flaccid myelitis [5,6]. EV-A71 has been associated with hand, foot and mouth disease and concurrent fatal encephalitis in very young children [7,8]. Echovirus 30 (E-30) is an emerging EV genotype, identified in most EV infections and often reported as responsible for meningitis [9,10].

PV, classified within species C EVs, is the causative agent of Poliomyelitis (polio), a severe disease that affects mostly children under five years of age. The most common initial symptoms of polio are fever, headache, malaise, or aseptic meningitis. Although these symptoms usually last 2–10 days and most cases recover completely, PV infection can cause permanent paralysis of the limbs and, in a few cases, fatal breathing muscles immobility [11]. In 1988, the World Health Assembly adopted a resolution for the worldwide eradication of polio, marking the launch of the Global Polio Eradication Initiative (GPEI). Since then, the number of paralytic polio cases has fallen by over 99%, from an estimated 350,000 to 140 reported cases in 2020 [12]. In June 2002, all WHO European Region Member States were certified as polio-free by the Regional Commission for the Certification of Poliomyelitis Eradication. High routine infant immunization coverage and acute flaccid paralysis (AFP) surveillance are part of the GPEI strategy and are considered essential components of the eradication plans. The number of AFP cases reported each year is used as an indicator of a country’s ability to detect polio, even in countries where PVs do not circulate. A country’s surveillance system needs to be sensitive enough to detect at least one case of AFP for every 100,000 children under 15 years of age [13]. Despite the usefulness and validity of AFP surveillance, many non-endemic countries have found it difficult to maintain high-sensitivity AFP surveillance in the absence of circulating PV and have adopted supplemental surveillance strategies to detect possible PV importations. Additional surveillance strategies and data have been accepted from countries with a long history of non-endemicity, high sanitation level, and strong health systems. These strategies include combinations of surveillance for polio cases and for cases of vaccine-associated paralytic poliomyelitis, environmental, and enterovirus surveillance [14]. Thanks to the GPEI, important milestones have been achieved in the fight against polio: wild poliovirus type 2 was eradicated in 1999 and no case of wild poliovirus type 3 has been reported since the last reported case in Nigeria in November 2012 [15,16]. Both type 2 and 3 wild type PV have officially been certified as globally eradicated. As of 2020, wild poliovirus type 1 affected two countries: Afghanistan and Pakistan [17]. Despite the success in achieving polio-free certification, some countries in the WHO European Region remain at high risk of possible polio outbreaks following importation, due, for instance, to a low population vaccine coverage and/or suboptimal surveillance performance, as demonstrated by the detection of two cases of polio due to circulating Vaccine-Derived Poliovirus (cVDPV) type 1 in Ukraine and the isolation in Bosnia and Herzegovina of one Sabin-like poliovirus type 2 strains, close to a VDPV [18,19]. The WHO Regional Reference Laboratory (RRL) at Istituto Superiore di Sanità (ISS) in Rome provides technical and scientific support for AFP surveillance activities in Serbia, Bosnia and Herzegovina, Montenegro, Bulgaria, Kosovo, Albania, North Macedonia, Malta, and Greece, and coordinates, together with the Italian Ministry of Health, the AFP surveillance in Italy. The RRL typically performs identification and characterization of polioviruses and other enteroviruses (EVs) from clinical specimens of AFP cases, provides reference materials, and trains personnel involved in laboratory-based surveillances activities. Hereby, six years of AFP surveillance data for 10 countries, including Italy, polio vaccination coverage, and EVs identification and typing, are reported.

## 2. Materials and Methods

### 2.1. Study Setting 

Retrospective analysis of data collected in the period January 2015–December 2020, on the AFP cases reported by Italy, Serbia, Bosnia and Herzegovina, Montenegro, Bulgaria, Kosovo, Albania, North Macedonia, Malta, and Greece was performed by the RRL. 

Data on AFP indicators and laboratory performance were obtained by the Centralized Information System for Infectious Diseases (CISID) database (https://data.euro.who.int/cisid/?TabID=543012 (accessed on 23 December 2021)). 

Data on countries’ vaccine coverage from 2015 to 2019, the number of samples analyzed in supplementary surveillance activities, and EVs isolation and typing data were obtained by the online Laboratory Data Management System (LDMS) of WHO/Europe (https://ldms.euro.who.int/Account/LogOn?ReturnUrl=%2f (accessed on 23 December 2021)).

### 2.2. AFP Surveillance and Performance Indicators

AFP surveillance is considered, by the WHO, to be the gold standard for detecting polio cases. According to the WHO clinical case definition, any child under 15 years of age with AFP or any person of any age with paralytic illness, if polio is suspected, has to be investigated [13]. WHO recommendation provides that all AFP cases should be reported immediately and investigated within 48 h and that two stool specimens should be collected 24–48 h apart and within 14 days of the onset of paralysis. A country’s surveillance system is expected to detect at least one case of AFP per 100,000 children under 15 years of age and it is recommended that a minimum 80% proportion of AFP cases have adequate stool specimens [13]. An AFP case has adequate samples if two stool specimens of sufficient quantity (8–10 g) are collected 24–48 h within 14 days of paralysis onset, and sent to the reference laboratory in good conditions [13]. 

According to the standard surveillance parameters recommended by the WHO, data on the number of AFP cases notified, on the annualized rate per 100,000 children under the age of 15, on the percentage of adequate stool specimens collected, on the surveillance index (SI) corresponding to a non-polio AFP rate up to 1.0 × (percentage of adequate specimen), and on the age groups of cases were analyzed for each country. 

### 2.3. Vaccine Coverage, Supplementary Surveillances, EVs Identification and Typing

Data on vaccine coverage in each country has been collected. Three doses of polio vaccine [oral polio vaccine (OPV) or inactivated polio vaccine (IPV)] administrated among children aged 12–23 months, for each year, were considered to determine the annual percentage of vaccine coverage for each country from 2015 to 2019. For each country, the presence of a supplementary surveillance, as EV and/or environmental surveillance, was verified, and the number of samples analyzed was also reported. 

EV isolation, identification, and typing were performed by each country according to WHO protocols [20,21,22]. Data reported in the LDMS database were analyzed by number and relative percentage of EVs identified and typed by country and surveillance type.

## 3. Results

### 3.1. Indicators of AFP Surveillance and Laboratory Performance

Table 1 reports data from six years of AFP surveillance data from 10 countries linked to the RRL in Rome. A total of 554 AFP cases were notified between 2015 and 2020 and for two thirds of them stool samples were collected within 14 days after symptoms’ onset. Malta and Kosovo reported only one AFP case from 2015 to 2020. In 2016 the highest number of AFP cases (N = 134) was recorded, with five countries achieving the WHO target of surveillance index (≥0.8). In 2019 and 2020, the number of countries that obtained an acceptable surveillance index decreased from two to zero, respectively. In the year 2020, the lowest number of AFP cases was reported (N = 56). Out of 554 notified AFP cases, 479 were children aged 0–15 years. As shown in Figure 1, more than 50% of AFP cases belonged to the 1–3 years and 4–6 years age groups.

### 3.2. Polio Vaccination Coverage

Figure 2 shows the percentage of children aged 12–23 months who received three doses of polio vaccine (OPV or IPV). Italy, Albania, Greece, Malta, Serbia, and Kosovo reported, at least since 2017, polio vaccination coverage ≥ 95%. Suboptimal coverage was reported in Bosnia and Herzegovina, Bulgaria, Montenegro, and North Macedonia.

### 3.3. Supplementary Surveillance Systems

Five countries performed supplementary surveillance activities in addition to AFP surveillance. As shown in Table 2, Italy and Malta analyzed environmental samples to monitor a potential circulation of poliovirus. Greece performed both enterovirus and environmental surveillance in 2018 and 2019. Enterovirus surveillance was constantly performed by Serbia from 2015 to 2020 and by Albania from 2015 to 2019. Italy and Albania were the countries that analyzed the majority of samples for environmental, and for enteroviruses surveillances, respectively (Italy, N = 1386; Albania, N = 555).

### 3.4. EV Isolation and Typing

Sabin-like poliovirus type 2 and 3 were isolated during AFP surveillance in 2017 and 2016, as already described. In particular, one Sabin-like poliovirus type 3 was isolated in 2017 from a child resident in Albania and two Sabin-like poliovirus type 2 were found in two children in Bosnia and Herzegovina in 2016 [19]. 

Two Sabin-like PV type 3 were isolated in 2015 and one in 2017 from sewage samples collected within the environmental surveillance in Italy. 

A total of 794 non-polio enteroviruses (NPEVs) were identified in samples collected through AFP, enterovirus and environmental surveillance. Most of the NPEVs were identified in Italy through environmental surveillance (N = 587; 17.08%) followed by those isolated in Albania (N = 41; 1.19%), Greece (N = 79; 2.30%), and Serbia (N = 27; 0.79%) during enterovirus surveillance activities. Fewer NPEVs were isolated in the context of AFP surveillance (Figure 3).

Out of 794 NPEVs identified, 243 were molecularly typed. A total of 22 viral strains were typed and isolated through AFP surveillance activities, while 72 and 149 were typed using enterovirus and environmental surveillance, respectively (Table 3). In sewage samples collected in Italy through environmental surveillance, the most frequently detected NPEVs were human echovirus 7 (N = 36), human coxsackievirus B5 (N = 32), and human coxsackievirus B4 (N = 17). Among the strains isolated within the enterovirus surveillance, most of them were typed as human coxsackievirus A24 (Greece; N = 21) and human echovirus 11 (Serbia; N = 9). Four human enterovirus 71 strains, known to cause epidemics of severe neurological disease and hand, foot and mouth disease in children [7,8], were isolated in Greece in two AFP cases.

## 4. Discussion

The aim of the AFP surveillance system is to promptly detect any polio suspected case and to adopt the adequate control measures to prevent wild poliovirus (WPV) and cVDPV spread. AFP surveillance indicators are used to monitor the quality of the surveillance activities and the target value is defined by the WHO to evaluate the surveillance performance of each country. The SI indicator combines the non-polio AFP rate (WHO target value ≥1) and the percentage of adequate stool specimens (WHO target value ≥ 80%) and represents one of the most significant performance parameters. The WHO has set the SI value ≥ 0.8 as a target to define an adequate surveillance performance. 

Considering the overall SI reported in this analysis, 18/60 values reached the WHO target. In 2019, only two countries, Bulgaria and North Macedonia, reported a surveillance index ≥ 0.8 and in 2020 none of the countries reached the target value. The low values of SI are mainly due to low values of non-polio AFP rate rather than due to the percentage of adequate stool specimens. In fact, while in more than half of percentage adequate specimens data reached the WHO target, there were only 11/60 AFP rate values ≥ 0.1. Underreporting of AFP cases can be explained by a change in perceived health priorities and decreasing attention on polio, due to the long-term absence of circulating polioviruses. Indeed, the last case of polio in this list of countries occurred in three children in Bulgaria in 2001, where WPV1 circulation was reported in susceptible children living in a low vaccination coverage area [23]. As shown by the number and the relative percentage of samples analyzed by type of surveillance and countries, Albania and Italy have adopted enterovirus and environmental surveillances with 555 and 1386 samples analyzed in the period, respectively. In 2020, the COVID-19 emergency further contributed to the decrease of AFP reported cases in all of the countries. In fact, as shown by a study on the impact of the COVID-19 pandemic on global poliovirus surveillance, the number of reported AFP cases declined by 33% and the mean number of days between the second stool specimen collection and receipt by the laboratory increased up to 70% [24]. Our data reflect the negative impact of the pandemic emergency in the routine AFP surveillance: in 2020 all analyzed countries reported an AFP rate < 0.8. Our findings also showed that the majority of AFP cases (52%) were in age groups under 5 years. This is in line with findings from previous studies in Kenya, Ghana, Iran, Nigeria, and India [25,26,27].

Thanks to polio vaccines, the burden of disease has been strongly reduced over the years: since 1988, wild polio cases have dropped by 99.9%, wild PV types 2 and 3 have been eradicated, and type 1 is currently endemic in only two countries—Afghanistan and Pakistan [17]. Since 2016, the trivalent OPV, containing all three types of Sabin strains, was replaced by bivalent OPV (bOPV; containing types 1 and 3 Sabin strains) and injectable inactivated poliovirus vaccine (IPV), containing antigens for all three PV types have been used in routine immunization programs worldwide [28]. Despite the efforts and results obtained by the GPEI, the presence of a cohort of unimmunized people led to an increase in the number of WPV1 cases in Afghanistan and Pakistan and cVDPV type 2 outbreaks.

Our vaccination coverage data highlighted some critical aspects: four countries (Bosnia and Herzegovina, Bulgaria, Montenegro, and North Macedonia) showed immunization under the reference value of 95% during the study period. In addition, it is very likely that the coverage values have dropped even more in 2020, in all countries, due to the COVID-19 pandemic. 

Our data suggest that some countries are at a high risk of polio outbreak following a possible importation of WPV or emergence of cVDPV, due to a low immunity of the population [29]. Furthermore, as previously reported by the WHO, several middle-income countries remain at intermediate risk of polio transmission, as their vaccination coverage appears to be declining and due to the suboptimal quality of poliovirus surveillance [30].

In Italy, the vaccination coverage was enhanced by the introduction, in 2017, of the National Immunization Plan (https://www.trovanorme.salute.gov.it/norme/dettaglioAtto?id=60201 (accessed on 23 December 2021)) that increased the number of mandatory vaccinations [31]. 

Environmental and enterovirus surveillance activities are adopted by many countries in support of the GPEI. However, although five countries reported data related to additional surveillance, only three reported systematic activities, suggesting that more efforts are needed in the implementation of other surveillance systems, which can provide an alert for a possible reintroduction of poliovirus. An infected person, even if they are asymptomatic, sheds large amounts of virus daily into the wastewater system for several weeks to months after infection [32]. Since PVs can persist in the environment for long periods, sewage screening can detect PV circulating in the community without clinical cases reported, especially in areas with high vaccination coverage with inactivated IPV. The usefulness of the environmental surveillance, as support to the AFP surveillance, has been demonstrated by several studies where PV was detected in the environment despite the absence of clinical cases in the population [33,34,35,36]. Another tool in support of the GPEI strategies is the enterovirus surveillance [14]. In the post-polio eradication era, enterovirus surveillance may provide an early warning system for a possible poliovirus reintroduction and for the detection and response to outbreaks of other potentially severe EVs, such as D68 and EV71.

Regarding the virus identification and typing data, no WPVs were isolated during the period. However, the isolation of two Sabin-like poliovirus type 2 strains in Bosnia and Herzegovina, one of them close to a VDPV, and the finding of a PV Sabin-like 3 virus in a child in Albania, requires particular attention. The genetic instability of Sabin strains permits occasional reversion to neurovirulence causing VAPP in vaccine recipients and infection in their unvaccinated contacts [37]. A low immunization coverage increases the risk of vaccine strains transmission leading to acquisition of new genetic variants resulting in the generation of new cVDPV [38].

Sabin-like PV type 3 was isolated in sewage samples of North Italy in 2015 and 2017. Since IPV completely replaced the OPV in Italy for safety reasons and risk-benefit considerations, the vaccine strains findings indicate the presence of people from regions in the world where OPV vaccination is still being used. 

NPEVs identification data showed that a high proportion of them were detected in the environment when compared with AFP and enterovirus surveillances. These data reflect the biological characteristics of EV infections. EVs are transmitted from person-to-person through the fecal-oral and oral-oral routes and are shed into the environment, where they can remain infectious for a long period of time. It was estimated that the amount of EV excreted into stools can reach the maximal amounts of 10^7^ infectious doses/day per person [32]. For these reasons, environmental and, in particular, sewage sites represent biological collectors where EVs are present in abundance. On the contrary, although the screening of AFP cases is the gold standard to highlight a possible PV circulation and enterovirus surveillance represents a valid support to PV eradication activities, there are two main reasons for the few numbers of NPEVs isolated: the paralysis can be frequently caused by non-infectious diseases, such as Guillain-Barre Syndrome, and the enterovirus surveillance is based on clinical signs and symptoms common to numerous viral and bacterial etiological agents.

NPEV typing data suggest a heterogeneous presence of viral strains: over the years numerous genotypes of coxsackievirus and echovirus have been isolated in all types of surveillance. However, our study has several limitations. The lacking of systematic and homogeneous enterovirus and environmental surveillance systems in the analyzed countries, and the small number of isolated and typed NPEVs in AFP and enterovirus surveillance, does not allow us to draw conclusions about the predominant genotypes. As reported by previous studies, typing of environmental NPEVs isolated in Italy showed the predominant circulation of echovirus 7, coxsackievirus B5, and coxsackievirus B4 [39,40]. Coxsackievirus, echovirus, and other EVs, such as D68 and EV71, cause severe diseases, particularly among young people, and represent a public health issue in industrialized countries. Environmental surveillance may provide an early detection system for human enteric pathogens and the isolated and typed virus may help in depicting the predominant circulating strains. 

## 5. Conclusions

In conclusion, our study describes the main results from surveillance activities in different countries aimed at maintaining the polio-free status. The monitoring of the real burden of EV diseases, including polio, is crucial to maintain the polio-free status. Efficient surveillance systems and high vaccination coverage are the two pillars to avoid polio reintroduction and spreading, and to respond to VDPV outbreaks in a timely manner.

## Figures and Tables

**Figure 1 vaccines-10-00044-f001:**
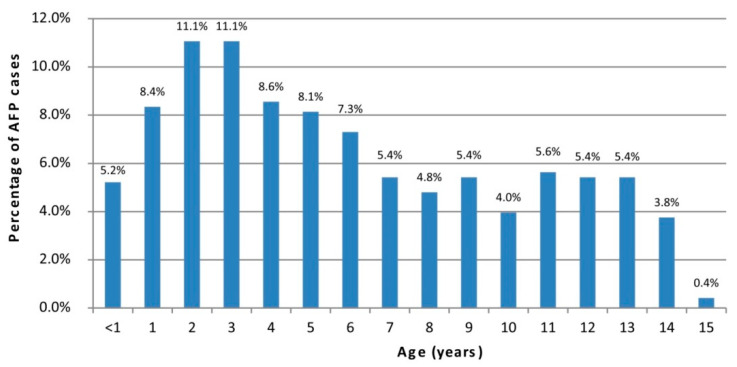
Percentage of AFP cases by age.

**Figure 2 vaccines-10-00044-f002:**
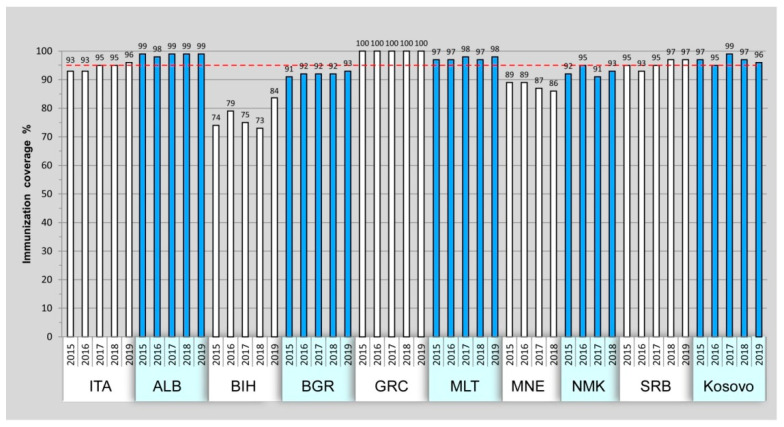
Three doses of polio vaccination coverage from 2015 to 2019 among children aged 12–23 months. Reference value of 95% coverage is shown by red line. Montenegro and North Macedonia lack data from 2019.

**Figure 3 vaccines-10-00044-f003:**
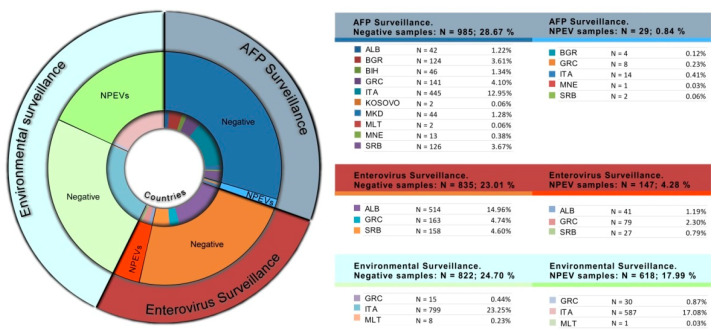
Number and relative percentage of NPEVs identified by type of surveillance and country, 2015–2020.

**Table 1 vaccines-10-00044-t001:** Six years of AFP surveillance data from Italy, Serbia, Bosnia and Herzegovina, Montenegro, Bulgaria, Kosovo, Albania, North Macedonia, Malta, and Greece.

Year	Inicators	Albania	Bosnia and Herzegovina	Bulgaria	Greece	Italy	Malta	Montenegro	Serbia	North Macedonia	Kosovo	TOTAL
**2015**	**No. of AFP Cases**	5	3	12	18	44	0	1	11	2	0	96.00
	**non-Polio AFP rate ***	0.77	0.6	**1.13**	**1.05**	0.51	0	0.86	0.65	0.59	0	0.65
	**% with 2 stool spec. ****	**100%**	67%	**100%**	**94%**	70%	N.A.	**100%**	73%	**100%**	N.A.	88.00%
	**Surveillance index *****	0.77	0.4	**1**	**0.94**	0.36	0	**0.86**	0.47	0.59	0	0.53
**2016**	**No. of AFP Cases**	8	10	17	18	68	0	2	10	3	0	134.00
	**non-Polio AFP rate ***	**1.25**	**2.27**	**1.6**	**1.05**	0.79	0	**1.72**	0.48	0.9	0	0.90
	**% with 2 stool spec. ****	**100%**	**100%**	**100%**	**100%**	69%	N.A.	100%	**90%**	**100%**	N.A.	94.14%
	**Surveillance index *****	**1**	**1**	**1**	**1**	0.54	0	**1**	0.43	**0.9**	0	0.77
**2017**	**No. of AFP Cases**	7	5	5	10	49	0	2	10	3	1	92.00
	**non-Polio AFP rate ***	**1.12**	**1.26**	0.47	0.58	0.57	0	**1.74**	0.6	0.91	2	0.62
	**% with 2 stool spec. ****	**100%**	**100%**	60%	**100%**	67%	N.A.	**100%**	30%	**100%**	100%	82.13%
	**Surveillance index *****	**1**	**1**	0.28	0.58	0.38	0	**1**	0.18	**0.91**	**1**	0.45
**2018**	**No. of AFP Cases**	2	1	10	18	39	0	1	8	3	0	82.00
	**non-Polio AFP rate ***	0.32	0.43	0.94	1.1	0.45	0	0.88	0.49	0.92	0	0.55
	**% with 2 stool spec. ****	**100%**	**100%**	**90%**	**89%**	67%	N.A.	**100%**	**88%**	**100%**	N.A.	91.75%
	**Surveillance index *****	0.32	0.43	**0.84**	**0.89**	0.3	0	**0.88**	0.43	**0.92**	0	0.44
**2019**	**No. of AFP Cases**	0	2	12	11	53	1	0	12	3	0	94.00
	**non-Polio AFP rate ***	0	0.43	**1.12**	0.69	0.62	**1.59**	0	0.74	0.93	0	0.63
	**% with 2 stool spec. ****	N.A.	**100%**	**100%**	**100%**	68%	100%	0%	**100%**	**100%**	N.A.	94.67%
	**Surveillance index *****	0	0.43	**1**	0.69	0.42	**1**	0	0.74	**0.93**	0	0.52
**2020**	**No. of AFP Cases**	2	1	6	10	31	0	0	5	1	0	56.00
	**non-Polio AFP rate ***	0.33	0.22	0	0.63	0.27	0	0	0.06	0.31	0	0.38
	**% with 2 stool spec. ****	**100%**	**100%**	**100%**	50%	52%	N.A.	0%	60%	**100%**	N.A.	80.29%
	**Surveillance index *****	0.33	0.22	0	0.32	0.14	0	0	0.04	0.31	0	0.23

* Annualized rate per 100,000 children under the age of 15. **Bold** = meeting WHO target of 1.0. ** Two stool specimens collected at least 24 h apart within 14 days of onset of paralysis and adequately shipped to the laboratory. **Bold** = meeting WHO target of 80%. *** Surveillance Index = non-polio AFP rate (not to exceed the target) × percent adequate stool collection. **Bold** = 0.8.

**Table 2 vaccines-10-00044-t002:** Number of samples analyzed by supplementary surveillance activities per country, 2015–2020.

Year	Italy	Albania	Greece		Malta	Serbia
	Environmental	Enterovirus	Enterovirus	Environmental	Environmental	Enterovirus
2015	206	95				17
2016	194	119				44
2017	231	104	88	8	3	17
2018	359	102	151	15		30
2019	198	135	3	22	3	75
2020	198				3	2
Total	1386	555	242	45	9	185

**Table 3 vaccines-10-00044-t003:** Number and percentage of typed NPEVs by surveillance and countries.

Surveillance (Number of NPEVs Typed)	Genotypes by Countries	Number of Genotypes	Percentage of Genotypes by Countries
AFP (N = 22)	BGR	2	
	Human_echovirus_30	2	100.00%
	GRC	8	
	Human_coxsackievirus_A16	2	25.00%
	Human_echovirus_18	2	25.00%
	Human_enterovirus_71	4	50.00%
	ITA	10	
	Human_coxsackievirus_B4	2	20.00%
	Human_echovirus_11	2	20.00%
	Human_echovirus_25	2	20.00%
	Human_echovirus_6	4	40.00%
	SRB	2	
	Human_coxsackievirus_B1-6	1	50.00%
	Human_echovirus_21	1	50.00%
Enterovirus (N = 72)	GRC	63	
	Human_coxsackievirus_A1	1	1.59%
	Human_coxsackievirus_A8	2	3.17%
	Human_coxsackievirus_A11	8	12.70%
	Human_coxsackievirus_A16	5	7.94%
	Human_coxsackievirus_A17	2	3.17%
	Human_coxsackievirus_A19	4	6.35%
	Human_coxsackievirus_A20	5	7.94%
	Human_coxsackievirus_A24	21	33.33%
	Human_coxsackievirus_B5	2	3.17%
	Human_echovirus_21	2	3.17%
	Human_echovirus_25	2	3.17%
	Human_echovirus_29	2	3.17%
	Human_echovirus_30	2	3.17%
	Human_enterovirus_A76	1	1.59%
	Human_enterovirus_B81	1	1.59%
	Human_enterovirus_C	3	4.76%
	SRB	9	
	Human_echovirus_11	9	100.00%
Environmental (N = 149)	GRC	17	
	Human_coxsackievirus_A11	2	11.76%
	Human_coxsackievirus_A13	4	23.53%
	Human_coxsackievirus_A24	4	23.53%
	Human_echovirus_11	3	17.65%
	Human_echovirus_30	3	17.65%
	Human_echovirus_9	1	5.88%
	ITA	132	
	Human_coxsackievirus_A4	1	0.76%
	Human_coxsackievirus_A5	2	1.52%
	Human_coxsackievirus_B2	1	0.76%
	Human_coxsackievirus_B3	5	3.79%
	Human_coxsackievirus_B4	17	12.88%
	Human_coxsackievirus_B5	32	24.24%
	Human_echovirus_1	6	4.55%
	Human_echovirus_11	9	6.82%
	Human_echovirus_12	1	0.76%
	Human_echovirus_13	1	0.76%
	Human_echovirus_14	3	2.27%
	Human_echovirus_18	1	0.76%
	Human_echovirus_19	1	0.76%
	Human_echovirus_3	3	2.27%
	Human_echovirus_30	2	1.52%
	Human_echovirus_6	11	8.33%
	Human_echovirus_7	36	27.27%
TOTAL		243	

## Data Availability

Not applicable.

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
