# Peer review of "Retrospective Analysis of Six Years of Acute Flaccid Paralysis Surveillance and Polio Vaccine Coverage Reported by Italy, Serbia, Bosnia and Herzegovina, Montenegro, Bulgaria, Kosovo, Albania, North Macedonia, Malta, and Greece"

_vaccines, 2021, doi:10.3390/vaccines10010044_

Round 1

Reviewer 1 Report

The authors presented in this study the  results of   the acute flaccid paralysis surveillance from 2015 to 2020  of the countries  linked to the WHO RRL at Istituto Suoeriore di Sanita , Italy. The vaccine coverage from 2015 to 2019EV isolation, identification and typing were recorded by each country according to WHO protocols.The environmental and  the enterovirus surveillance were  used as supplement to the AFP surveillance providing  the required information about a country to document maintenance of its polio-free status

The results are very interesting , giving us a map of the enterovirus circulation in different  geographic areas.   

In conclusion the  efficient surveillance systems and high vaccination coverage are mandatory to maintain the polio-free status, to decrease the risk of polio reintroduction and spreading in according with the WHO recommendation. 

Author Response

Comments and Suggestions for Authors

The authors presented in this study the  results of   the acute flaccid paralysis surveillance from 2015 to 2020  of the countries  linked to the WHO RRL at Istituto Suoeriore di Sanita , Italy. The vaccine coverage from 2015 to 2019, EV isolation, identification and typing were recorded by each country according to WHO protocols.The environmental and  the enterovirus surveillance were  used as supplement to the AFP surveillance providing  the required information about a country to document maintenance of its polio-free status. 

The results are very interesting , giving us a map of the enterovirus circulation in different  geographic areas.   

In conclusion the  efficient surveillance systems and high vaccination coverage are mandatory to maintain the polio-free status, to decrease the risk of polio reintroduction and spreading in according with the WHO recommendation. 

- We thank  the reviewer for this comment.

Reviewer 2 Report

Thanks to the authors for addressing this important topic. 

Line 128, I think there is a typo

I think it is better to revise the long title of figure 2.

It is good to add the main limitations of the study.

Author Response

Comments and Suggestions for Authors

Thanks to the authors for addressing this important topic. 

1) Line 128, I think there is a typo

- “Considering” was changed in “according to”.

2) I think it is better to revise the long title of figure 2.

- We changed the figure 2 title. Countries codes were removed from the title.

3) It is good to add the main limitations of the study.

- We thank  for this comment. Limitations of the study is added in the  revised version.

Reviewer 3 Report

This can be potentially a good and interesting paper. But there is still much to be done.

1) The introduction is very confusing. Is this paper about polio only or also including other enteroviruses. There is very little mention of the other non-polio viruses in the introduction.

2) What are the differences between poliovirus and other enteroviruses in terms of genetics, taxonomy and clinical manifestation of paralysis? There is no description of these anywhere in the paper.

3)Did the authors attempt to find correlation (r) between percentage of paralysis and rate of vaccination?

4)" enterovirus surveillance does not allow draw conclusions about the predomi-323nant genotypes" need to be corrected to:

" enterovirus surveillance does not allow US TO draw conclusions about the predomi-323nant genotypes"

Author Response

Comments and Suggestions for Authors

This can be potentially a good and interesting paper. But there is still much to be done.

  • The introduction is very confusing. Is this paper about polio only or also including other enteroviruses. There is very little mention of the other non-polio viruses in the introduction.

  • What are the differences between poliovirus and other enteroviruses in terms of genetics, taxonomy and clinical manifestation of paralysis? There is no description of these anywhere in the paper.

Points 1-2. We thank  for these comments. As every type of enterovirus surveillance is generally adopted to support the polio eradication initiative, the main focus of introduction was on poliovirus. However, we do agree  with the observation  about a little mention of the other non-polio enteroviruses in the introduction: a description of NPEVs genetics and taxonomy characteristics  and their clinical manifestation is added in the Introduction of revised version.

3) Did the authors attempt to find correlation (r) between percentage of paralysis and rate of vaccination?

- AFP surveillance showed, except for 3 cases out 554 that received Oral Polio Vaccine (OPV) few weeks before the collection of the samples, (see section 3.4. EV isolation and typing Ref. 19), the absence of cases correlated to poliovirus circulation . So we didn’t attempt to find a correlation between non-polio paralysis and poliovirus vaccination.

4)"enterovirus surveillance does not allow draw conclusions about the predominant genotypes" need to be corrected to:

" enterovirus surveillance does not allow US TO draw conclusions about the predominant genotypes"

- We changed, accordingly.

Round 2

Reviewer 3 Report

Improvements seen